# Changing trends of corporate social responsibility reporting in the world-leading airlines

**Lu Yang** *, **Cindy S. B. Ngai**, **Wenze Lu**

Department of Chinese and Bilingual Studies, The Hong Kong Polytechnic University, Kowloon, Hong Kong SAR, China

* jasminelu.yang@connect.polyu.hk

## Abstract

The recent outbreak of major economic, environmental and social incidents (e.g. privacy leakage, corruption) has put the blooming airline industry under spotlight. However, scant studies examined the trends and changes of corporate social responsibility (CSR) practices through CSR reporting on the economic, environmental and social dimensions in the aviation industry. To fill the gap, the current study examines the economic, environmental and social themes of Global Reporting Initiative (GRI) Standards on CSR communication of the global airlines. Quantitative content analysis is adopted for analyzing and comparing the CSR themes in 105 CSR reports of 21 airlines in major leagues from 2013 to 2017. Our findings indicated a salient growing trend in CSR reporting in the economic dimension while GRI 201-Economic Performance was the most significantly reported sub-topic. Our study has also compared the CSR reporting practices between the APEC based airlines and EU based airlines. The results indicated that focus of CSR reporting was different in the two regions. This study provided a comprehensive understanding of how global airlines communicate their CSR practices and respond to stakeholders' expectations via CSR reporting. This study informs both academics and practitioners about the changing trends of CSR development and reporting in the airline industry worldwide.

**Data Availability Statement:** All relevant data are within the manuscript and its Supporting Information files.

**Funding:** The authors received no specific funding for this work.

## Introduction

With the development of tourism, air travel has been a popular option among travelers. It is an indispensable part of the tourism and transportation sector [1]. The aviation industry has seen dramatic growth over the last two decades. According to the statistics from the International Civil Aviation Organization (ICAO), the number of globally carried passengers has increased from 1.4 billion in 1998 to 3.9 billion in 2017 [2] and reached 4 billion in 2018 [3]. Global prosperity depends on air connectivity, as air-travel is a basic means of transport "as well as a driver of global economic, social and cultural development" [4] (p.1). The airline industry has extensive economic implications on society. It facilitates logistics and transportation, promotes investment in city facilities and events, and most importantly, creates jobs [5]. 8.4 million jobs

**Competing interests:** The authors have declared that no competing interests exist.

were directly offered by the air transport sector [4]. Besides, air transport plays an essential role in the world's greatest industry—Travel and Tourism. It caters specialized training and significant education opportunities to the young generation. Nonetheless, the negative impacts of airline industry on the natural environment (e.g., biodiversity loss, noise pollution, gaseous emissions, climate change, water quality, and waste production), as well as the social and economic issues brought by air transport are significant [6–8]. In recent years, the outbreak of social and economic issues in major airlines, including fatal accidents in 2014, mistreating Asian passengers in 2017, leakage of customers' information in 2018, and corruption scandals in 2018, violated the United Nations' Global Compact [9] and the social and economic standards (e.g., GRI-406 Non-discrimination, GRI-418 Customer privacy, GRI-416 Customer Health & Safety, GRI-205 Anti-Corruption) in Global Reporting Initiative [10]. These issues have wreaked global havoc on the airline industry, which also raised public awareness of airlines' CSR performance. It is hardly surprising that the airline sector has been in the forefront of the debate over CSR performance.

In response to the introduction of guidelines from United Nations Global Compact [9] and the Rio+20 International Civil Aviation Organization briefing in 2012, service providers in the airline sector have been required to step up on their CSR practices to minimize the negative impacts on the environment [6] and bring more positive influences on society. Apart from the extrinsic factors, corporate social responsibility is also crucial to the sustainable development of airlines given its impact on stakeholders' engagement and corporate branding [11]. Stakeholders' positive perception of the airlines' CSR practices can contribute to the increase of stakeholders' loyalty and trust [12]. Thus, many airlines have put efforts into the adoption of CSR initiatives and made "sustainable aviation" become a commonplace practice [13]. Furthermore, the CSR effort does not only include adopting CSR practices, also involve communicating CSR initiatives to their stakeholders [14, 15]. Subsequently, CSR reporting, an efficient way to present the CSR efforts, commitments and achievements of the organizations and engage stakeholders [1] plays an important role in CSR initiative communication [16]. To date, CSR studies on the aviation industry mainly focused on case studies or regional studies. Any detailed and large-scale investigation into airlines' CSR reporting of major CSR dimensions viz. economic, environment, and social, is absent. To fill this gap, this longitudinal study aims to reveal the latest trend of CSR reporting on the economic, environment, and social dimensions in the airline industry worldwide.

This paper is structured as follows. First, we reviewed the literature on CSR, its importance, and CSR reporting in the aviation industry, followed by research questions and hypothesis. Then we explained the research method employed in addressing the research questions and hypothesis. Finally, we outlined findings, discussion, implications together with limitations of this study.

## Theoretical background

### The definition of CSR and its importance in the airline industry

Corporate Social Responsibility is evolving and has developed into an intricate concept [17]. It took up different definitions in different eras. Back in the 1950s, research [18] defined CSR as societal obligation of business to follow policies, lines, make decisions, and take responsible action in accordance with the goals and values of the society. Later on, CSR study began to emphasize on the importance of stakeholders' benefits, on top of stockholders' interest [19]. At the turn of millennium, the notion of CSR was discussed at the World Business Council for Sustainable Development, where CSR is defined as "the continuing commitment by business to behave ethically and contribute to economic development while improving the quality of

life of the workforce and their families as well as the local community and society at large" [20] (p.3). The council further claimed that the three main dimensions of sustainable development: economy, environment, and society, constitute an integral part of CSR. Researchers have further suggested that CSR involves voluntary activities of companies that exerted positive impacts on the society, beyond the interests of the company and which is required by law [21]. Recently, the European Commission [22] defined CSR as the responsibility of firms for their influences on society. It is generally agreed that the basic idea of CSR is to assess how corporations engage with their stakeholders by integrating social values with stakeholders' interests and fulfill the obligations of sustainable societal development.

Previous study [23] suggested that corporate sustainability depends on the its social legitimacy, economy and environmental performance. As air-travel has inevitably influenced the social, natural, and economic environment, there is an increasing stakeholder expectation for the airline industry to reduce their negative effects by adopting appropriate CSR practices [24]. CSR is an intangible asset in the aviation sector. It brings benefits to this industry in several aspects. An affirmative connection between financial performance and CSR investment was noticed [25]. Furthermore, financial analysts and fund managers believe that investment in social responsibilities would bring firms interests [26]. It has been suggested that an efficient CSR policy of an airline is a key factor in higher stock return and investor satisfaction [11], which would lead to positive profitability [27], firm value presentation [28], and higher financial performance [29].

CSR practices are closely associated with companies' competitiveness and strategic survival. It is generally believed that airline companies are voluntarily engaging themselves with CSR activities to enhance the competitive advantage [30] and distinguish themselves from other competitors [31]. It is worth noting that CSR plays an important role in the development of long-term business strategies and values [32] as it contributes to the reduction of risk, employee commitment and improvement of corporate reputation [33].

Research confirmed that CSR practices have positive impact on airlines' reputation [34] and customers' loyalty [35]. Stronger relationships can be built between corporations and their stakeholders via CSR practices and the effective CSR communication [36]. A study [37] confirmed that an effective CSR performance has a significant impact on corporate image, corporate branding, customer equity, market share, and it can lead to positive customer attitude and behavior [38,39]. Other studies [40–44] suggested that customer identification, support, satisfaction and loyalty, in particular, are positively associated with efficient implementation of CSR. To be specific, customers are willing to patronize products and services from companies that perform ethical behavior towards their stakeholders and consequently involve customer themselves in moral social causes [45]. In addition, a favorable CSR perception by customers might alleviated the negative impacts, especially when the service failure was attributed to a stable cause [12, 46].

## Challenges of CSR reporting in the airline industry

Implementing CSR does not solely consist of CSR effort; how companies present their CSR policies and practices to stakeholders is also important [15, 47]. Most companies communicate with their stakeholders on CSR initiatives implementation through corporate websites, media releases, voluntary communications or corporate sustainability reports [6]. Typically, CSR reports were regarded as marketing instruments [48] and efficient communication tool to enrich stakeholders' understanding of companies' commitments and actions [49,50]. The quantity and quality of CSR reports have gradually improved over the past two decades [51,52]. Many corporations have changed their one-page declaration of environmental and

social responsibilities into more comprehensive CSR/sustainability reports or annual reports with detailed information on specific plans and outcomes [8]. However, the reporting practice varies significantly in terms of disciplines, scope and depth [53].

Prior study [54] suggested that the production of high-quality CSR reports yields both internal and external benefits. Internally speaking, CSR reports provide comprehensive tracking of corporate sustainability performance and emphasize the link between financial and non-financial performance which leave a positive effect on staff engagement and work efficiency [55]. Externally, corporations see the CSR reports as a means to express corporate's effort in mitigating negative environmental, social and governance influences and a way to improve reputation and brand loyalty. It also enhances stakeholders' understanding of the organization's value [55].

Though previous studies confirmed the importance of CSR practice and reporting in sustainable corporate development [56], the number of airlines that has officially and consistently producing CSR reports was relatively small compared to the size of the industry [6]. The airline industry did not put sufficient effort in green reporting until the agenda 21 which encourages organizations "to report annually on their environmental records as well as on their use of energy and natural resources" was put forward in the Rio's Earth Summit 1992 [57] (p.5). In fact, CSR reporting practice varies among airlines, especially when there is a lack of legal and regulatory compliance at the turn of the century. With the recent development of airline CSR practice and communication, the quantity and quality of CSR reports were greatly improved. Enhancement in the breadth and depth of CSR reports is witnessed over the last twenty years [58]. For instance, British Airway introduced the concept of sustainability into its report at 2000 [8]. At the same time, other airlines, such as Air France, Cathay Pacific, Delta, KLM, Lufthansa, and SAS started to recognize the significance of the "triple bottom line" of environmental, social and economic responsibilities. Besides, some leading service carriers have begun to apply the generic guidelines or frameworks for their CSR reporting together with the validating statistical summaries. Although the system of CSR reporting has developed in the past few decades, some drawbacks, such as low credibility [59], using unstable results [60], lack of consistency [6, 61] were found in the reporting process.

## The need for researching into CSR reporting in aviation industry

In recent years, the significance of CSR communication for the tourism industry has been acknowledged [62]. Nonetheless, limited attention has been paid to researching into CSR practices and the reporting of the global aviation industry [63]. As mentioned in the introduction, CSR effort does not only include CSR practices, but also CSR reporting [14, 15] for communicating the CSR efforts, commitments and achievements of the organizations and engage stakeholders [1]. Early studies on CSR reporting in the aviation industry indicated that the airline industry focused heavily on environmental issues than economic or social effects [6, 28, 63]. A previous study has also revealed that some of the airlines reported CSR initiatives following the legal and regulatory guidelines, while others chose not to [6]. Recently, a change of CSR reporting practice is noted where airlines are found to report more on social and economic responsibilities [24, 61, 64]. To fully reveal the changing landscape of CSR reporting in the aviation industry, a longitudinal study on the reporting of major CSR issues is highly warranted.

In addition, the existing CSR research in the aviation industry remains mostly anecdotal or case studies, focusing on CSR practice in a specific airline or a particular country/region [63, 65]. For instance, there were studies investigating the airlines' environmental reporting in Japan [66], Asia-Pacific [67], and Europe [68]. A handful of studies compared the airlines' CSR practices between regions. Some studies reported that Western based airlines exhibited

remarkable CSR practices, whereas Asia-Pacific airlines performed poorly and need improvement [1]. In particular, European airlines were found to have provided more information on community issues, sustainable development, and employee involvement while most of the Asia-based airlines provided less information on community and labour issues [35]. A more comprehensive and up-to-date view on how global airlines practice CSR communication is highly warranted. To provide a comprehensive view of CSR communication and reporting in the airline industry, this study is going to conduct a 5-year longitudinal analysis on 105 CSR reports of 21 global airlines from 2013 to 2017 to uncover the development and changes of CSR reporting in the airline industry.

## Research questions and hypothesis

The first research question is proposed to reveal the trend and changes of CSR reporting along the three major CSR dimensions namely economic, environment and social issues in the airline industry [6, 24, 28, 61, 63–64] from 2013 to 2017.

RQ1: What was the trend of CSR reporting in the economic, environment and social dimensions in the airline industry from 2013 to 2017? Were there any changes on the reporting of sub-topics over time?

To further examine the focus of the CSR reporting practices in the global airlines, we inquired into the significant differences between the sub-topics of the three dimensions in the CSR reports from 2013 to 2017 and put forward the following research question:

RQ2: Were there significant differences in the reporting of sub-topics of the three topic-specific standards from 2013–2017?

Though economic, environmental and social issues are the major CSR dimensions reported in airline industry, previous studies unveiled a shift in the focal point of CSR reporting. Recent findings from CSR reporting studies indicated that both airlines [24, 61] and airport operators [64] have placed more attention on reporting economic and social issues than environmental issue. Therefore, the following hypothesis was formulated:

H1. Airlines addressed more on economic and social issues than environmental issues in their CSR reports.

In addition, previous research [35] witnessed differences between Europe-based airlines and Asia-Pacific based airlines in their CSR communication. In this study, by categorizing the majority of the selected airlines into Asia-Pacific Economic Cooperation (APEC) based airlines and European Union (EU) based airlines, we examined the different CSR reporting practices between the two major operating regions. Hence, the last research question is as follows:

RQ3: What were the differences between APEC based airlines and EU based airlines in CSR communication? Were there any changes over time?

## Methodology

### Sampling and data collection

We sampled the airlines from the three major world airline alliances viz. Star Alliance, Skyteam, and Oneworld which account for 82% of international market share in the aviation industry [69]. Members of the three alliances are from different parts of the world in which a

majority of them are the flag-carriers that lead the innovation of the industry and set the trend of development [6].

Since the study investigates the development of CSR communication through analyzing the official CSR reports of air carriers, we only selected the airlines that have published the reports online with open access. Those did not release the reports accessible or available on the internet platforms (e.g., GRI database, official homepages, etc.), as well as those communicate selected sustainability information on their websites without producing reports in the downloadable format, were excluded. Furthermore, to conduct a longitudinal analysis, we selected airlines which published a report every year during the sampled period. Airlines released sporadic reports were excluded. As a lingua franca, English is the most widely employed language of communication with stakeholders in the aviation industry. Therefore, we have only examined the reports produced in English. Given the merging of Air France and Royal Dutch Airlines (KLM) into the Air France-KLM Group in 2004, we count them as one company. In total, 21 airlines were selected.

Next, we decided the sampling period. Since the importance of the balance between the three pillars of sustainability: economic, environmental and social responsibilities was reinforced in the Rio+20 International Civil Aviation Organization briefing held in 2012 [4], we would like to examine the progress and development of CSR implementation in the aviation industry, especially on its reporting practices. Therefore, we chose to sample reports from 2013 onwards. A previous study [70] suggested a good longitudinal study requires at least three times of repeated measurements. Consequently, a five-year longitudinal study was adopted in this research.

Lastly, we collected the airlines' reports that communicated the companies' CSR practices [66]. Given the wide range of report titles found (e.g. *CSR Report*, *Sustainability Report*, *Annual Report*, *Corporate Responsibility Report*, *Sustainable Development Report*, *Environmental and Social Responsibility Report*, *Environmental Performance Report*), we decided to include those reports that focus on corporate CSR practices. As a result, we collected 105 CSR reports (comprising of 4,662,021 words) of 21 airlines from 2013 to 2017. (See S1 Appendix for the selected airlines, the reporting year and word numbers of the respective CSR report).

## Content analysis

Content analysis is a widely used method for researchers to conduct objective, systematic, and quantitative analyses on content communication in CSR research [15, 71]. This study has also employed quantitative content analysis method to address the research questions and test the hypothesis.

We have determined to adopt a CSR reporting framework as the structural framework for coding in content analysis. However, the question remains which CSR framework is most appropriate in the evaluation of CSR reporting. It has been discussed by some studies [1, 6] that ISO14000, ISO26000, SA8000, and Global Reporting Initiative (GRI) Standards are adopted by airlines when preparing the CSR reports. These guidelines are the basic frameworks of CSR reporting which present a clear examination of sustainable development. Based on the descriptions of the International Organization for Standardization's (ISO) official website [72], ISO14000 is a standard related to environmental management that mainly focuses on how to minimize the negative impact on the environment during operation while ISO26000 emphasizes social responsibility practices and focuses on corporate governance and stakeholder issues. Human rights, labor issues, and social welfare are the primary objectives of SA8000, whereas, GRI Standards provides the most comprehensively coverage on environmental, economic, and social impacts. GRI framework of sustainability reporting assists

**Table 1. Topic-specific standards and sub-topics adopted from GRI standards.**

| GRI 200 series: Economic | GRI 300 series: Environmental | GRI 400 series: Social |
|---|---|---|
| GRI 201: Economic Performance<br>GRI 202: Market Presence<br>GRI 203: Indirect Economic Impacts<br>GRI 204: Procurement Practices<br>GRI 205: Anti-Corruption<br>GRI 206: Anti-competitive Behavior | GRI 301: Materials<br>GRI 302: Energy<br>GRI 303: Water and Effluents<br>GRI 304: Biodiversity<br>GRI 305: Emissions<br>GRI 306: Effluents and Waste<br>GRI 307: Environmental Compliance<br>GRI 308: Supplier Environmental Assessment | GRI 401: Employment<br>GRI 402: Labor/Management Relations<br>GRI 403: Occupational Health and Safety<br>GRI 404: Training and Education<br>GRI 405: Diversity and Equal Opportunity<br>GRI 406: Non-discrimination<br>GRI 407: Freedom of Association and Collective Bargaining<br>GRI 408: Child Labor<br>GRI 409: Forced or Compulsory Labor<br>GRI 410: Security Practices<br>GRI 411: Rights of Indigenous Peoples<br>GRI 412: Human Rights Assessment<br>GRI 413: Local Communities<br>GRI 414: Supplier Social Assessment<br>GRI 415: Public Policy<br>GRI 416: Customer Health and Safety<br>GRI 417: Marketing and Labeling<br>GRI 418: Customer Privacy<br>GRI 419: Socioeconomic Compliance |

organizations to identify, gather and report in a transparent and comparable manner. Recently, there has been a growing use of GRI framework in CSR reporting in the aviation industry [6]. Consequently, we have selected the three topic-specific GRI standards of the 2016 version as our coding scheme. These standards are categorized into three series: GRI 200 (Economic topics), GRI 300 (Environmental topics), and GRI 400 (Social topics). Moreover, each topic-specific standard contains sub-topics specific to that standard (see Table 1 for all the sub-topics to be included in the Economic, Environment and Social standards). In sum, 33 sub-topics in the GRI Standards [10] were employed as the framework of coding.

To conduct coding in the reports collected, we identified the keywords in each sub-topic using the word frequency function in Nvivo10, a powerful data analysis computer software provides rich description of data [73]. Keyword is the smallest unit which contains the most important information and represents the meaning of the entire text [74]. Next, we filtered out the common keywords (such as *standards*, *guidelines*, *organizations*, *topic*, *reporting*, *etc.*) in all the sub-topics. To avoid inflating the intensity by including overlapping keywords in the sub-topics [75], the first and second authors re-examined the word frequency list to remove the overlapping keywords. Subsequently, only the top 6–9 topic- specific keywords with a weighting 0.34 or above in each GRI sub-topic were selected to avoid compromising the balance among the sub-topics. (See S2–S4 Appendices for the specific keywords in each GRI sub-topic).

Next, we identified and coded the intensity of each topic-specific keyword in the 33 GRI sub-topics in each CSR report. For example, the word "energy" is one of the recognized keywords of the GRI 302 standard. We found the count of "energy" in the word frequency of Air Canada 2013 CSR report was 28, and 34 in the 2014 CSR report. Hence, we marked 28 and 34 as the intensity of the word "energy" under the Air Canada 2013 and 2014 report, respectively. Furthermore, if the selected keyword is abbreviation, we count the intensity in both full form and abbreviation. For instance, GHG standards for *Greenhouse Gas*, we coded both *Greenhouse Gas* and GHG in CSR reports. Since the total word count of each report was not the same, to reflect the proportion of the keyword usage in reports, we standardized the keyword intensity counts by dividing the total keyword count of each sub-topic by the total word count of each report. For instance, the total word count of the 2016 CSR report of Air Canada was

23,988 and the intensity of keywords in GRI 401 was 128. Then, the standardized intensity of keywords of Air Canada's 2016 report was 0.53% in GRI 401.

## Statistical analysis

To address RQ1, we calculated the annual growth rate (AGR), the average annual growth rate (AAGR) of the adoption of each sub-topic, median of AGR, and median absolute deviation (MAD) to identify the changing trend of CSR reporting in four periods (2013–2017). As for RQ2 and H1, we performed one-way ANOVA to analyze the standardized keyword counts to reveal their variances and mean differences of use in the GRI Standards of the CSR reports. Since heteroscedasticity of data is quite common in real world dataset, we have conducted the test of homogeneity of variances to see if this assumption is violated. We would replace the ANOVA results with those of Welch ANOVA if the assumption of homogeneity of variances was violated [76]. To address RQ3, we calculated AAGR to compare the reporting on GRI sub-topics between the two regions throughout the five-year period.

## Results

### The changing trend of CSR communication in the airline industry

RQ 1 inquired the changing trends of CSR reporting in the airline industry. The annual growth rate and average annual growth rate (AAGR) of 33 sub-topics between 2013 and 2017 is examined to determine the changes and trend. Table 2 indicated that the highest AAGR of reporting is witnessed on GRI 418 (7.73%), followed by GRI 404 (7.30%) and GRI 307 (7.18%). Whereas the lowest AAGR is found in GRI 205 (-1.81%), GRI 302 (-1.47%), GRI 410 (-1.10%).

As reported in Fig 1A, a positive annual growth rate (AGR) in GRI 201, GRI 203, GRI 206 in 2013–2014 was noted. Also, we found a positive AGR in all economic sub-topics in the 2014–2015. In the reporting year of 2015–2016, a positive AGR was observed in GRI 202, GRI 203 and GRI 205 while a positive AGR in GRI 201, GRI 204, GRI 206 was found in the reporting year of 2016–2017. Furthermore, a salient growing trend on GRI 201, GRI 204, and GRI 206 reporting was witnessed in the last period.

Regarding the CSR reporting in environmental dimension (Fig 1B), positive AGR in GRI 301, GRI 304, GRI 306, GRI 307 was found in 2013–2014, while the positive AGR in GRI 302, GRI 303, GRI 304,GRI 306, GRI 307, and GRI 308 was noted in 2014–2015. However, only two sub-topics (GRI 302, GRI 305) in 2015–2016 and 6 sub-topics (GRI 301, GRI 303, GRI 304, GRI 305, GRI 306, and GRI 307) in 2016–2017 recorded a positive AGR. In comparison with the third period, a growing trend of reporting on GRI 301, GRI 303, GRI 304, GRI 305, GRI306, and GRI 307 were found in the last period.

As for the social dimension (Fig 1C), we observed positive AGR in GRI 401, GRI 404, GRI 405, GRI 406, GRI 407, GRI 411, GRI 418, and GRI 419 in 2013–2014 whereas positive AGR of all social sub-topics was witnessed in 2014–2015. As for 2015–2016, positive AGR of 7 sub-topics (GRI 406, GRI 407, GRI 408, GRI 409, GRI 410, GRI 413, and GRI 414) was found while positive AGR in 16 sub-topics (except for GRI 409, GRI 410, GRI 413) were uncovered in 2016–2017. In comparison with the third period, a growing trend of reporting on GRI 401, GRI 402, GRI 403, GRI 404, GRI 405, GRI 407, GRI 408, GRI 411, GRI 412, GRI 414, GRI 415, GRI 416, GRI 417, GRI 418, and GRI 419 was observed in the fourth period.

### The adoption of topic specific GRI Standards in the airline industry

RQ2 examined if there were significant differences on CSR reporting of the sub-topics in three dimensions. For the economic dimension (Fig 2A), a statistically significant difference

**Table 2. The summary of average annual growth rate, median and median absolute deviation in each sub-topic over the five-year period.**

| Dimensions | Sub-dimensions | Average Annual Growth Rate | Median | Median Absolute Deviation |
|---|---|---|---|---|
| Economic Dimension | GRI 201 | 5.66% | 5.20% | 4.53% |
| | GRI 202 | -0.65% | -1.38% | 4.53% |
| | GRI 203 | 4.64% | 2.13% | 2.78% |
| | GRI 204 | 3.10% | 2.35% | 4.79% |
| | GRI 205 | -1.81% | 1.32% | 2.53% |
| | GRI 206 | 4.83% | 4.98% | 3.30% |
| Environmental Dimension | GRI 301 | 2.96% | 1.30% | 2.42% |
| | GRI 302 | -1.47% | -0.39% | 3.25% |
| | GRI 303 | 2.56% | 1.01% | 3.79% |
| | GRI 304 | 5.44% | 7.17% | 4.31% |
| | GRI 305 | 3.65% | -3.67% | 14.67% |
| | GRI 306 | 1.80% | 1.98% | 1.77% |
| | GRI 307 | 7.18% | 9.14% | 5.25% |
| | GRI 308 | -0.57% | -0.03% | 4.61% |
| Social Dimension | GRI 401 | 4.86% | 5.76% | 2.19% |
| | GRI 402 | 2.93% | 1.65% | 5.20% |
| | GRI 403 | 1.79% | 0.14% | 2.66% |
| | GRI 404 | 7.30% | 7.57% | 4.22% |
| | GRI 405 | 4.62% | 5.78% | 1.45% |
| | GRI 406 | 3.03% | 2.91% | 1.59% |
| | GRI 407 | 4.70% | 3.50% | 2.86% |
| | GRI 408 | 4.54% | 4.81% | 2.48% |
| | GRI 409 | 2.76% | 2.46% | 4.36% |
| | GRI 410 | -1.10% | -1.69% | 5.29% |
| | GRI 411 | 4.26% | 5.08% | 1.31% |
| | GRI 412 | 2.44% | 2.17% | 3.96% |
| | GRI 413 | 0.11% | -0.61% | 6.47% |
| | GRI 414 | 0.07% | 0.81% | 1.26% |
| | GRI 415 | 2.43% | 0.36% | 2.57% |
| | GRI 416 | 1.58% | 1.60% | 6.46% |
| | GRI 417 | 4.70% | 4.98% | 5.75% |
| | GRI 418 | 7.73% | 7.82% | 6.95% |
| | GRI 419 | 3.30% | 2.64% | 2.45% |

between groups existed ($F(5,624) = 216.395$, $p<0.0001$****). The post hoc Tukey results indicated that the intensity of GRI 201 reporting (M = 0.464, SD = 0.32) was significantly higher than the remaining sub-topics (GRI 202 (M = 0.005, SD = 0.003, $p<0.0001$****), GRI 203 (M = 0.003, SD = 0.001, $p<0.0001$****), GRI 204 (M = 0.006, SD = 0.004, $p<0.0001$****), GRI 205 (M = 0.004, SD = 0.002, $p<0.0001$****), and GRI 206 (M = 0.008, SD = 0.003, $p<0.0001$****)). No significant differences were found among the other sub-topics. In other words, GRI 201 was the most intensively reported sub-topic in the economic dimension.

As to the environmental dimension, a statistically significant difference between groups was observed ($F(7,832) = 26.244$, $p<0.0001$****). The post hoc Tukey test showed there are statistically significant differences between GRI 301 (M = 0.009, SD = 0.004) and GRI 302 (M = 0.006, SD = 0.003, $p<0.0001$****), GRI 304 (M = 0.006, SD = 0.003, $p<0.0001$****), GRI 307 (M = 0.005, SD = 0.003, $p<0.0001$****), GRI 308 (M = 0.004, SD = 0.003, $p<0.0001$****), GRI 303 (M = 0.007, SD = 0.004, $p = 0.012$*), and GRI 305 (M = 0.012, SD = 0.012,

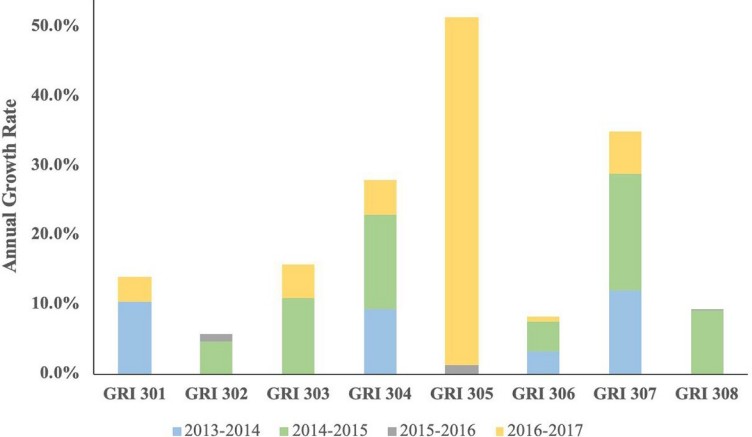

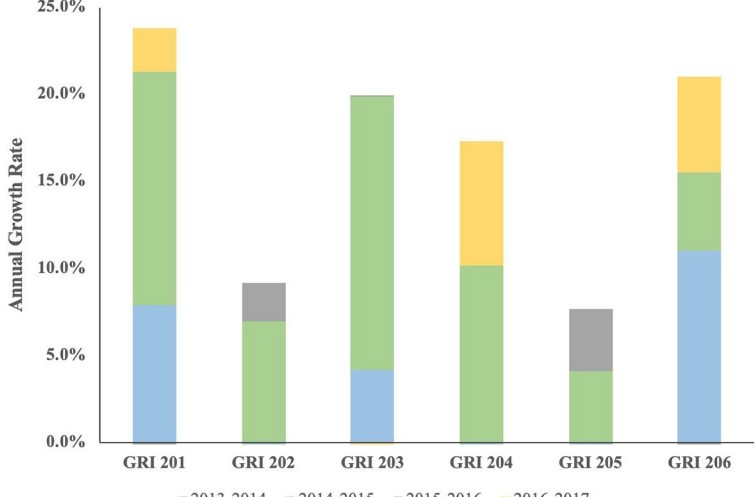

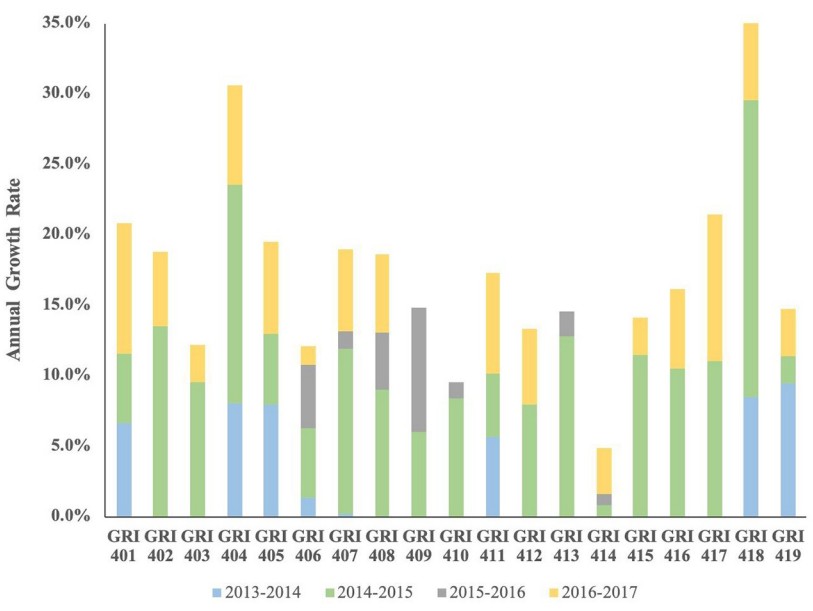

**Fig 1.** A) Trends of CSR reporting in the economic dimension in the airline industry. B) Trends of CSR reporting in the environmental dimension in the airline industry. C) Trends of CSR reporting in the social dimension in the airline industry.

$p = 0.003^{**}$). A high significant difference was witnessed between GRI 302 (M = 0.006, SD = 0.003) and GRI 305 (M = 0.012, SD = 0.012, $p<0.0001^{****}$), while a strong significant difference was found between GRI 302 (M = 0.006, SD = 0.003) and GRI 306 (M = 0.009, SD = 0.004, $p = 0.001^{***}$).

Besides, the intensity of reporting on GRI 305 (M = 0.012, SD = 0.012) was strongly higher than GRI 306 (M = 0.009, SD = 0.004, $p = 0.001^{***}$) and significantly higher than GRI 303 (M = 0.007, SD = 0.004, $p<0.0001^{****}$), GRI 304 (M = 0.006, SD = 0.003, $p<0.0001^{****}$), GRI 307 (M = 0.005, SD = 0.003, $p<0.0001^{****}$) and GRI 308 (M = 0.004, SD = 0.003, $p<0.0001^{****}$). Same pattern was found in the reporting of GRI 306 (M = 0.009, SD = 0.004),

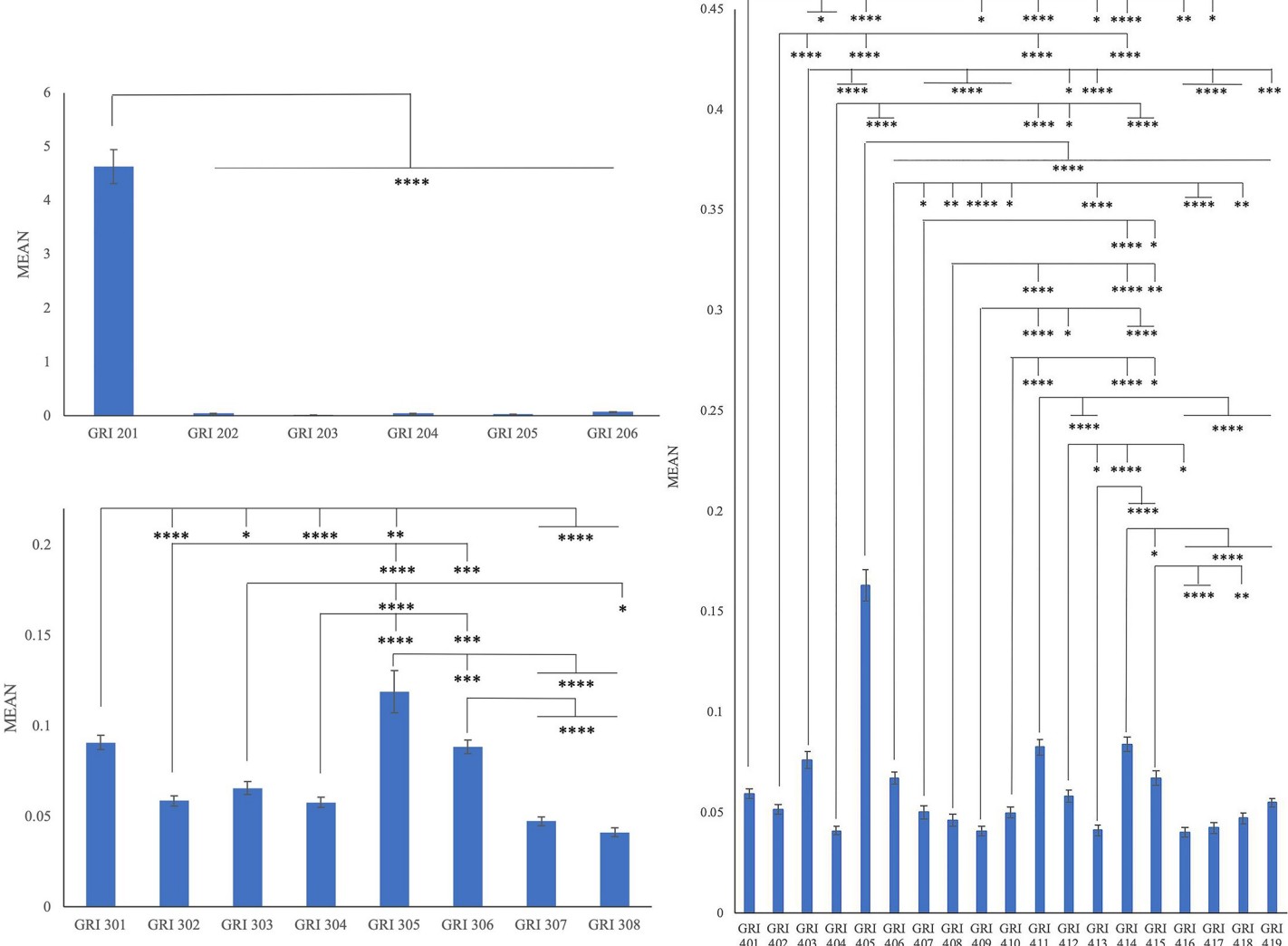

**Fig 2.** A) The intensity of reporting on sub-topics in economic dimension on CSR reporting over a period of 5 years. B) The intensity of reporting on sub-topics in environmental dimension on CSR reporting over a period of 5 years. C) The intensity of reporting on sub-topics in social dimension on CSR reporting over a period of 5 years.

which is more intensively reported than GRI 307 (M = 0.005, SD = 0.003, $p<0.0001$****) and GRI 308 (M = 0.004, SD = 0.003, $p<0.0001$****). A strong significant difference was revealed between GRI 306 (M = 0.009, SD = 0.004) and GRI 304 (M = 0.006, SD = 0.003, $p = 0.001$***). There was no significant difference between GRI 307 (M = 0.005, SD = 0.003) and GRI 308 (M = 0.004, SD = 0.003, $p = 0.99$). Furthermore, the highest intensity level of GRI 305 was found in airlines' CSR reports and the lowest intensity level of GRI 308 was observed. In sum, GRI 305 was the most adopted sub-topic (Fig 2B).

In terms of the social dimension, we witnessed a statistically significant difference between groups ($F$(18,1976) = 71.630, $p<0.0001$****). The post hoc Tukey test revealed that the intensity of reporting on GRI 405 (M = 0.016, SD = 0.008) was significantly higher than the remaining sub-topics (e.g., GRI 401 (M = 0.006, SD = 0.003, $p<0.0001$****), GRI 402 (M = 0.005, SD = 0.003, $p<0.0001$****), GRI 403 (M = 0.008, SD = 0.004, $p<0.0001$****), GRI 404 (M = 0.004, SD = 0.002, $p<0.0001$****), GRI 406 (M = 0.007, SD = 0.003, $p<0.0001$****), GRI 407 (M = 0.005, SD = 0.003, $p<0.0001$****) GRI 408 (M = 0.005, SD = 0.003, $p<0.0001$****), GRI 409 (M = 0.004, SD = 0.003, $p<0.0001$****), GRI 410 (M = 0.005, SD = 0.003, $p<0.0001$****), GRI 411 (M = 0.008, SD = 0.004, $p<0.0001$****), GRI 412 (M = 0.006, SD = 0.003, $p<0.0001$****), GRI 413 (M = 0.004, SD = 0.003, $p<0.0001$****), GRI 414 (M = 0.008, SD = 0.004, $p<0.0001$****), GRI 415 (M = 0.007, SD = 0.004, $p<0.0001$****), GRI 416 (M = 0.004, SD = 0.002, $p<0.0001$****), GRI 417 (M = 0.004, SD = 0.003, $p<0.0001$****), GRI 418 (M = 0.005, SD = 0.003, $p<0.0001$****), and GRI 419 (M = 0.006, SD = 0.002, $p<0.0001$****).

Same pattern was found in both GRI 411 (M = 0.008, SD = 0.004) and GRI 414 (M = 0.008, SD = 0.004). Highly significant differences can be witnessed in these two sub-topics between GRI 401 (M = 0.006, SD = 0.003, $p<0.0001$****), GRI 402 (M = 0.005, SD = 0.003, $p<0.0001$****), GRI 404 (M = 0.004, SD = 0.002, $p<0.0001$****), GRI 405 (M = 0.016, SD = 0.008, $p<0.0001$****), GRI 407 (M = 0.005, SD = 0.003, $p<0.0001$****), GRI 408 (M = 0.005, SD = 0.003, $p<0.0001$****), GRI 409 (M = 0.004, SD = 0.003, $p<0.0001$****), GRI 410 (M = 0.005, SD = 0.003, $p<0.0001$****), GRI 412 (M = 0.006, SD = 0.003, $p<0.0001$****), GRI 413 (M = 0.004, SD = 0.003, $p<0.0001$****), GRI 416 (M = 0.004, SD = 0.002, $p<0.0001$****), GRI 417 (M = 0.004, SD = 0.003, $p<0.0001$****), GRI 418 (M = 0.005, SD = 0.003, $p<0.0001$****), and GRI 419 (M = 0.006, SD = 0.002, $p<0.0001$****), respectively. As for the significant differences among the remaining sub-topics please refer to Fig 2C. Overall speaking, GRI 405 was the most reported sub-topic in the social dimension.

Hypothesis 1 assumed global airlines valued more economic and social issues than environmental issues on the CSR communication. There was a statistically significant difference between groups as determined by one-way ANOVA ($F$(2, 3462) = 172.317, $p<0.0001$****). A Tukey post hoc test revealed that the intensity of reporting on economic issues (M = 0.081, SD = 0.214) was statistically significantly higher than environmental issues (M = 0.007, SD = 0.006, $p<0.0001$****) and social issues (M = 0.006, SD = 0.004, $p<0.0001$****). However, no statistically significant difference between the environmental and social issues ($p = 0.963$) was noted. Thus, the H1 is partially supported.

## Comparison between the EU based airlines and APEC based airlines in CSR reporting

Research question 3 inquired about the differences and changes of CSR reporting among the airlines of the two major operating regions. The AAGR of each sub-topic reported by airlines in both regions from 2013 to 2017 was examined because of the different group sizes of the two regions (S5 Appendix). Though no significant difference was observed in the reporting of sub-topics between the EU and APEC airlines, the AAGR of 16 sub-topics of EU airlines over-took that of APEC airlines (Fig 3). On the other hand, the AAGR of 17 sub-topics of APEC

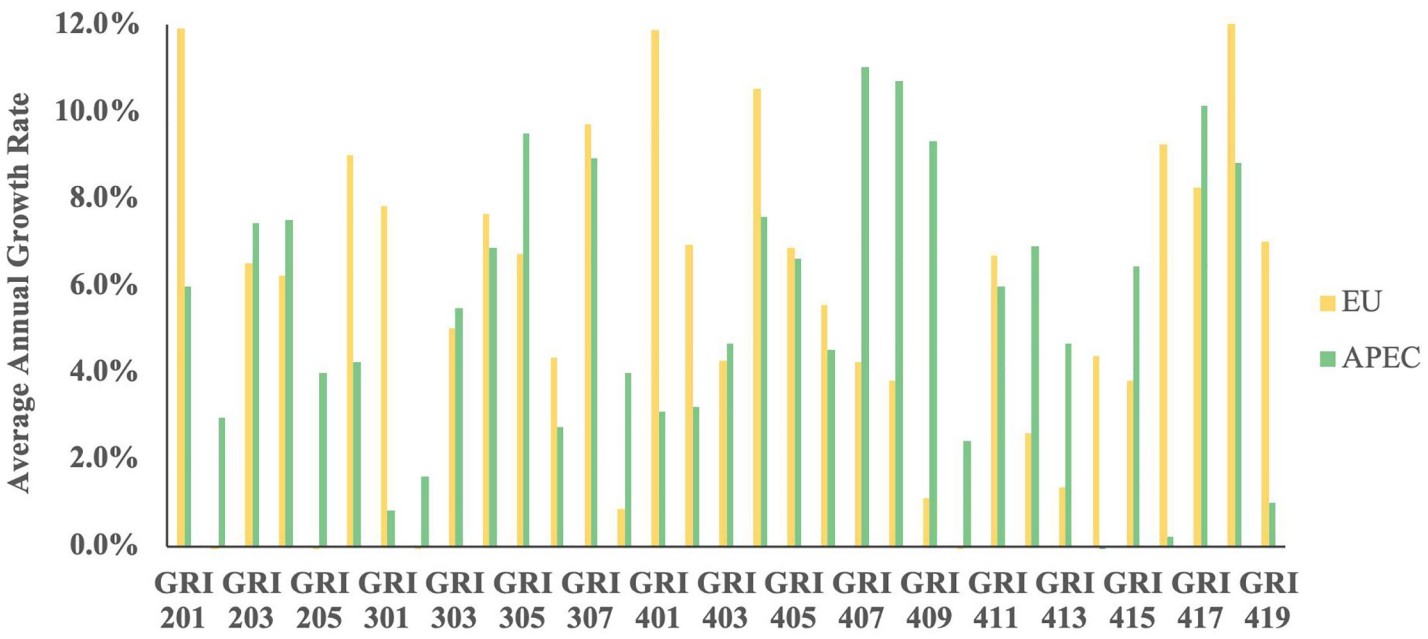

**Fig 3. The average annual growth rate of each sub-topics on CSR reporting among the airlines of the two major operating regions from 2013 to 2017.**

airlines is higher than that of EU airlines. In comparison with APEC airlines, the EU airlines reported significantly more on GRI 201, GRI 206, GRI 301, GRI 401, GRI 402, GRI 414, GRI 416, GRI 419 which are sub-topics related to balance the market competition, environmental compliance, employee management, and customer safety. While the APEC airlines reported more on the sub-topics related to energy consumption, human rights and local community development (e.g., GRI 202, GRI 205, GRI 302, GRI 407, GRI 408, GRI 409, GRI 410, GRI 412, GRI 413). In conclusion, APEC airlines have been exerting efforts on CSR reporting by showing positive AAGR in 32 sub-topics throughout the five years.

## Discussion

This paper provided an overview of the implementation of CSR practices among global airlines and the latest development of CSR communication through the CSR reports. Using quantitative content analysis to examine the 105 CSR reports of 21 global airlines from 2013 to 2017, we revealed the changing trends in CSR reporting in the airline industry.

### A salient growth on CSR reporting in the economic dimension in the airline industry

Congruent with recent studies that pointed out both airlines and airport operators are found to report more on economic responsibilities [24, 60, 64], our results confirmed that economic issues were significantly reported and intensively communicated with stakeholders than environmental, and social issues in the airline industry from 2013 to 2017. For the past decade, the airline industry is making noteworthy improvements across a range of sustainability matters, especially on the environmental aspect. the increased usage of "low-carbon technology, environmentally friendly materials, new aircraft systems, and sustainable energy sources" [4] (p.6) is witnessed in the carriers across various regions. These improvements might explain the low intensity of reporting on environmental issues as heightened enactment may not be

urgently required. Especially when there is a gradual decline of airfares and drastic increase in the cost of living, employees' wages, and operation costs, the net profit margin of global scheduled airlines has gradually decreased [4]. Airlines have to place more attention on the economic dimension instead of environmental issues to ensure the balanced development in the three dimensions of sustainability. Therefore, it is not surprising to note the heightened reporting on economic dimension, the GRI 201 *'Economic Performance'* in particular. Profitability making has become one of the biggest challenges in the aviation industry. Despite the aviation-friendly background generating significant profits for this industry, airlines earned less than six dollars per passenger [77]. Consequently, the air carriers had to make changes to enhance their economic performance.

## Reporting of labour rights in the social dimension might become the focal point in future CSR reporting in the aviation industry

It is noteworthy that we have found a promising and salient growth in the reporting of labour rights. A positive growth in the reporting of sub-topic of GRI 406 *'Non-discrimination'* and GRI 407 *'Freedom of Association and Collective Bargaining'* is noted from 2013–2017. Corroborated with findings from a previous study that analyzed 9,500 sustainability reports from all sectors, our findings also witnessed a steady increase in the CSR reporting on employee and labor practices of the social dimension in aviation industry [48]. This might be associated with the recent call for balanced sustainable development from the United Nations [9, 57]. Airlines are required to place more attention on the reporting of social dimensions in the coming years.

## APEC based airlines made significant progress in CSR communication and have caught up with EU airlines

As opposed to the previous studies [1, 35], our findings uncovered that the AAGR in 17 sub-topics reported by APEC airlines was higher than that of the EU airlines from 2013 to 2017, though the EU airlines still managed to have a higher AAGR in 16 sub-topics when compared with to the APEC counterparts. Apparently, APEC based airlines have enhanced their CSR reporting practices and made great efforts to catch up with their EU counterparts in CSR communication.

Though the Asia-Pacific countries just began to acknowledge and adopt the CSR practices decades ago [1], they are catching up quickly, especially with the rapid development of digital science and technology in the last two decades. Asian and South American countries have transformed their economic models and gradually strengthened the economic expansion to measure up to the developed countries. As the economies improve, people focus more on social and environmental issues and sustainability of their living conditions. As a result, both governments and the public in the APEC regions are more aware of sustainable development in their society. CSR practices have become one of the core advocacies in these countries. The airline industry is also called upon to participate in advancing sustainable development.

## Contribution and limitation

Air travel is a popular means of public transportation. As there are millions of public stakeholders in the industry, a favorable CSR reporting plays an important role in enhancing reputation of the airlines. Moreover, the outbreak of major incidents in recent years has put the industry under spotlight. Airlines have been required to ramp up their CSR practices to bring about more positive influences on the society. By investigating the CSR reporting of airline

industry from 2013–2017, this study sheds lights on the research and practice of CSR reporting and the findings could be of value to both academics and practitioners. Academically speaking, this study contributes to the body of literature on CSR reporting in the aviation industry by conducting a longitudinal analysis on 105 CSR reports of 21 global airlines from 2013 to 2017. The changes in CSR reporting, the new initiatives of CSR reporting and the salient growth in CSR reporting of the APEC airlines are revealed in the current study.

This study also offers practical value to industry professionals. By revealing the trend and reporting of the three major GRI dimensions in major airlines around the world, airlines and related businesses in the aviation industry could devise better CSR strategies and enhance their CSR reporting practices so as to better address the needs of their stakeholders and strengthen their reputation. By comparing CSR reporting practices between two regions, this study usefully informs CSR leaders and officers on their CSR policy making when communicating with stakeholders across different regions.

The first limitation of this study is the selected samples were full-service airlines and the majority of the airlines were flag carriers which assumed to be the leading corporations in the industry. The low-cost airlines were not included. There is a possibility that the adoption of CSR practices might not the same between these two types of airlines. Thus, it is necessary to mention that the results of this research would not represent the entire industry. Further studies can investigate the development of CSR communication among low-cost airlines. The second limitation is that we have not explored the CSR practices from the passengers' perspective. To examine if what the airlines valued aligns with what the passengers want, may cater important insights to the industry. Future studies can examine how passengers perceive the airlines' CSR communication and the degree of stakeholder engagement. Another constraint of this study is concerned with the use of GRI standards as the coding framework. This framework is designed for all the industries around the world and not tailor-made for the airline industry. A multi-framework study is suggested for future research in CSR reporting in the aviation industry context.

## Supporting information

**S1 File. The intensity of keywords in GRI 200 series among the 21 airlines.**
(XLSX)

**S2 File. The intensity of keywords in GRI 300 series among the 21 airlines.**
(XLSX)

**S3 File. The intensity of keywords in GRI 400 series among the 21 airlines.**
(XLSX)

**S1 Appendix. The selected airlines, reporting year and word count of the CSR reports.**
(DOCX)

**S2 Appendix. The selected keywords in the sub-topics of economic issues.**
(DOCX)

**S3 Appendix. The selected keywords in the sub-topics of environmental issues.**
(DOCX)

**S4 Appendix. The selected keywords in the sub-topics of social issues.**
(DOCX)

**S5 Appendix. The categorization of EU based airlines and APEC based airlines.**
(DOCX)

## Acknowledgments

We would like to thank Mr. Patrick Pak Kei Ng for his helpful advice on academic English. We would like to thank the Editor and the reviewers for their constructive criticism to help us immensely improve our original manuscript.

## Author Contributions

**Conceptualization:** Lu Yang, Cindy S. B. Ngai.

**Data curation:** Lu Yang, Cindy S. B. Ngai.

**Formal analysis:** Lu Yang.

**Methodology:** Cindy S. B. Ngai.

**Supervision:** Cindy S. B. Ngai.

**Visualization:** Lu Yang, Wenze Lu.

**Writing – original draft:** Lu Yang.

**Writing – review & editing:** Lu Yang, Cindy S. B. Ngai, Wenze Lu.

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
