## [Decision Letter · Decision Letter 0]

24 Apr 2020

PONE-D-20-09210

Changing trends of corporate social responsibility reporting in the world-leading airlines

PLOS ONE

Dear Ms. Yang,

Thank you for submitting your manuscript to PLOS ONE. After careful consideration, we feel that it has merit but does not fully meet PLOS ONE’s publication criteria as it currently stands. Therefore, we invite you to submit a revised version of the manuscript that addresses the points raised during the review process.

The manuscript should be extended in terms of motivation and contribution to the existing literature, discussion concerning prior studies, research methodology, as well as concluding remarks and policy implications.

We would appreciate receiving your revised manuscript by Jun 08 2020 11:59PM. To enhance the reproducibility of your results, we recommend that if applicable you deposit your laboratory protocols in protocols.io, where a protocol can be assigned its own identifier (DOI) such that it can be cited independently in the future. For instructions see: http://journals.plos.org/plosone/s/submission-guidelines#loc-laboratory-protocols

We look forward to receiving your revised manuscript.

Kind regards,

Stefan Cristian Gherghina, PhD. Habil.

Academic Editor

PLOS ONE

Reviewers' comments:

Reviewer's Responses to Questions

**Comments to the Author**

1. Is the manuscript technically sound, and do the data support the conclusions?

Reviewer #1: Partly

Reviewer #2: Partly

Reviewer #3: Partly

2. Has the statistical analysis been performed appropriately and rigorously? 

Reviewer #1: Yes

Reviewer #2: Yes

Reviewer #3: I Don't Know

3. Have the authors made all data underlying the findings in their manuscript fully available?

Reviewer #1: Yes

Reviewer #2: Yes

Reviewer #3: Yes

4. Is the manuscript presented in an intelligible fashion and written in standard English?

Reviewer #1: Yes

Reviewer #2: No

Reviewer #3: Yes

5. Review Comments to the Author

Reviewer #1: The purpose of the peer-reviewed article is to identify the structure of airline CSR reporting in dynamics (over several years). In particular, the authors identified and investigated several research questions:

1) the structure and dynamics of sub-topics of CSR reports;

2) to identify the proportional distribution of the information between topics in CSR reports;

3) to identify the difference in the structure of airlines' CSR reports in APEC and EU.

To answer the above questions, the authors applied the well-known method of content analysis.

In general, the application of the method and statistical analysis is understandable and performed rather thoroughly. However, the rationale for the problem of airline CSR reporting and the logic of the study is not clear enough. For example, several cases of violation of passenger rights and a corruption scandal in one of the airlines were highlighted. However, the airline CSR reporting problem has quite a few aspects and the reporting structure, which, in theory, should reflect the interests of all market actors, stakeholders is indeed still under development.

The value of the research conducted by the authors could be better understood if the Contribution section of the article covered in more detail the main conclusions, forecasts or suggestions of the authors, for example, on improving the structure of airline CSR reports.

The article should be carefully read and minor inaccuracies in English corrected.

Reviewer #2: I would like to thank you editor for asking me to review this paper. It focuses on a very interesting research question, especially today after the Covid 19 outbreak since aviation industry should widen their CSR agenda.

I have some comments

At the end of the introduction, the structure of the article should be presented, ie the sections that will follow.

There is confusion in the development section of work cases. While it seems that the authors are developing scientific questions, a working hypothesis that refers to another methodology is finally being submitted. Some explanations and clarifications should be given at this point and the work hypotheses that emerge from the relevant literature should be described more effectively.

Finally, the discussion should show the contribution of authors’ results in the field of aviation industry and sustainable development as well as to corporate sustainability reporting.

Put some relative literature in section of Hypothesis deelopment

Karagiannis, I., Vouros, P., Skouloudis, A., & Evangelinos, K. (2019). Sustainability reporting, materiality, and accountability assessment in the airport industry. Business Strategy and the Environment, 28(7), 1370-1405.

Tsalis, T. A., Botsaropoulou, V. D., & Nikolaou, I. E. (2018). A methodology to evaluate the disclosure practices of organisations related to climate change risks: a case study of international airports. International Journal of Global Warming, 15(3), 257-276.

Reviewer #3: Interesting article, well written, important topic, but with room for improvement:

- Review Abstract: it's too long and should stick to objective of the paper; Sampling; methodology; conclusions.

- Quick re-reading of the paper: minor typos and needs standardization of citations in the document - sometimes uses number and other times names;

- Methodological concerns:

1. needs additional details on how the authors chose keywords for content analysis;

2. Keywords are shared across sub-topics. Given the methodology leads to overlapping. Could you show the impact on test results if you clean any overlap? otherwise you are inflating intensity across sub-topics and it is more probable that you mis-identify reporting in each sub-topic.

3. why use intensity only for testing and nothing about binary identification of keywords? Plus, your measure of reporting is highly affected by wording (longer reports). In Psychology and Impression Management literature longer explanations are associated with obfuscation and misleading users. Could your results just show airlines pretending to provide more information? You should look for alternative measures across which your results are robust – relative frequency (%) of keywords; binary frequency of keywords, etc.

4. Given potential Heteroskedasticity, I recommend non-parametric testing, at least as robustness.

5. Averaging across firms increases the difficulty to spot outliers. Yet, in the literature about CSR, firms mimic each other in the same industry. Robust analysis should include how and in what sub-topics firms deviate from average reporting. This is for instance what you find in the literature as committed vs opportunistic reporting (ex. Gonçalves, T., Gaio, C., & Costa, E. (2020). Committed vs opportunistic corporate and social responsibility reporting. Journal of Business Research.)

- Motivation: Your motivation does not link with your research questions and analysis. If you want to check whether firms that had a scandal decide to engage more in CSR you should distinguish, in test, firms with a scandal in a given year and onward versus other firms.

- Conclusions and implications: Poor job in providing implications of the research, for management, policy and stakeholders. Finally, in terms of conclusions, a general remark: Why is it important to focus on airline industry? What is specific to this context? What can we generalize to other industries?

6. PLOS authors have the option to publish the peer review history of their article (what does this mean?). If published, this will include your full peer review and any attached files.

Reviewer #1: Yes: Dr Kristina Marintseva

Reviewer #2: No

Reviewer #3: No

---

## [Author Response · Author response to Decision Letter 0]

11 May 2020

The following are our responses to the constructive comments and suggestions made by the reviewers. All reviewers’ comments and suggestions are quoted verbatim.

Comments from Reviewer One:

The purpose of the peer-reviewed article is to identify the structure of airline CSR reporting in dynamics (over several years). In particular, the authors identified and investigated several research questions:

1) the structure and dynamics of sub-topics of CSR reports;

2) to identify the proportional distribution of the information between topics in CSR reports;

3) to identify the difference in the structure of airlines' CSR reports in APEC and EU.

To answer the above questions, the authors applied the well-known method of content analysis.

In general, the application of the method and statistical analysis is understandable and performed rather thoroughly. However, the rationale for the problem of airline CSR reporting and the logic of the study is not clear enough. For example, several cases of violation of passenger rights and a corruption scandal in one of the airlines were highlighted. However, the airline CSR reporting problem has quite a few aspects and the reporting structure, which, in theory, should reflect the interests of all market actors, stakeholders is indeed still under development.

>> Response: We thank the reviewer for the encouraging words and constructive feedback. Additional literature and explanation on problem of CSR reporting in the airline industry and the objective of this paper is added on page 4 in the Introduction and page 10 in the Theoretical Background.

The value of the research conducted by the authors could be better understood if the Contribution section of the article covered in more detail the main conclusions, forecasts or suggestions of the authors, for example, on improving the structure of airline CSR reports.

>> Response: We thank the reviewer for the insightful suggestion. We have rewritten the Contribution section with detailed examples to highlight the theoretical and practical implication of the study respectively (page 28-29).

The article should be carefully read and minor inaccuracies in English corrected.

>> Response: We express our heartfelt thanks to the reviewer for noting this. We are truly sorry for the mistakes made. We have invited an English native speaker who is also an academic editor to help edit and proofread the manuscript before resubmission.

Comments from Reviewer Two:

I would like to thank you editor for asking me to review this paper. It focuses on a very interesting research question, especially today after the Covid 19 outbreak since aviation industry should widen their CSR agenda.

I have some comments:

At the end of the introduction, the structure of the article should be presented, ie the sections that will follow.

>> Response: We thank the reviewer for the constructive feedback. We have added the structure of this paper on page 5 at the end of the Introduction.

There is confusion in the development section of work cases. While it seems that the authors are developing scientific questions, a working hypothesis that refers to another methodology is finally being submitted. Some explanations and clarifications should be given at this point and the work hypotheses that emerge from the relevant literature should be described more effectively.

Put some relative literature in section of Hypothesis deelopment

Karagiannis, I., Vouros, P., Skouloudis, A., & Evangelinos, K. (2019). Sustainability reporting, materiality, and accountability assessment in the airport industry. Business Strategy and the Environment, 28(7), 1370-1405.

Tsalis, T. A., Botsaropoulou, V. D., & Nikolaou, I. E. (2018). A methodology to evaluate the disclosure practices of organisations related to climate change risks: a case study of international airports. International Journal of Global Warming, 15(3), 257-276.

>> Response: We thank the reviewer for this invaluable suggestion. We found the paper titled “Sustainability reporting, materiality, and accountability assessment in the airport industry” very helpful. We have added this recommended paper in the Theoretical Background (page 10), Hypothesis development (page 11-12), and the Discussion section (page 26). In addition, we revised our Figures based on data visualization method in this paper (page 20-21, Fig 1a-c).

Finally, the discussion should show the contribution of authors’ results in the field of aviation industry and sustainable development as well as to corporate sustainability reporting.

>> Response: Once again, we are grateful to the reviewer for the constructive feedback. We have rewritten the Contribution section with detailed examples to highlight the theoretical and practical implications for sustainable development and CSR reporting in aviation industry (page 28-29).

Comments from Reviewer Three:

Interesting article, well written, important topic, but with room for improvement:

- Review Abstract: it's too long and should stick to objective of the paper; Sampling; methodology; conclusions.

>> Response: We are grateful to the reviewer for noting this. We have shortened and revised the Abstract.

- Quick re-reading of the paper: minor typos and needs standardization of citations in the document - sometimes uses number and other times names;

>> Response: We are truly sorry for the mistakes made. We have standardized the in-text citations. In addition, we have invited an English native speaker who is also an academic editor to help edit and proofread the manuscript before resubmission.

- Methodological concerns:

1. needs additional details on how the authors chose keywords for content analysis;

>> Response: We thank the reviewer for this useful suggestion. Further elaboration on the process of keywords selection is added on page 17 in the Method.

2. Keywords are shared across sub-topics. Given the methodology leads to overlapping. Could you show the impact on test results if you clean any overlap? otherwise you are inflating intensity across sub-topics and it is more probable that you mis-identify reporting in each sub-topic.

>> Response: We are grateful to the reviewer for this insightful feedback. We agreed that the overlapping keywords might inflate the measuring of intensity. In this resubmission, we have carefully re-examined the keywords generated and revised the keywords list by removing the overlapping keywords and re-matching the sub-topics with the appropriate keywords. Consequently, we have rebuilt the topic-specific keywords list (page 33-34, Appendix 2-4) and redone all coding in the sub-topics. All the findings in the Results section has been revised based on the new coding results (page 18-25).

3. why use intensity only for testing and nothing about binary identification of keywords? Plus, your measure of reporting is highly affected by wording (longer reports). In Psychology and Impression Management literature longer explanations are associated with obfuscation and misleading users. Could your results just show airlines pretending to provide more information? You should look for alternative measures across which your results are robust – relative frequency (%) of keywords; binary frequency of keywords, etc.

>> Response: Thanks again for this constructive suggestion. As we have selected the top 6-9 topic- specific keywords with a weighting 0.34 or above in each GRI sub-topic to avoid compromising the balance among the sub-topics, the measuring of intensity would allow us to explore further on the variance of reporting in different GRI sub-topics. Further information on the keywords selection and standardization is provided in the Method on page 17.

4. Given potential Heteroskedasticity, I recommend non-parametric testing, at least as robustness.

>> Response: We thank the reviewer for this invaluable suggestion. As we highly agree that the ANOVA results might be affected by heteroscedastic data, we have performed Levene’s test to see if the assumption of homogeneity of variances was violated. We would replace the ANOVA results with those of Welch ANOVA if the assumption of homogeneity of variances was violated. Additional literature is also added to provide further explanation on the statistical analysis in the Method (page 18).

5. Averaging across firms increases the difficulty to spot outliers. Yet, in the literature about CSR, firms mimic each other in the same industry. Robust analysis should include how and in what sub-topics firms deviate from average reporting. This is for instance what you find in the literature as committed vs opportunistic reporting (ex. Gonçalves, T., Gaio, C., & Costa, E. (2020). Committed vs opportunistic corporate and social responsibility reporting. Journal of Business Research.)

>> Response: We are grateful to the reviewer for noting this. We have revised our findings and presented the mean, mean average deviation, or median and median absolute deviation in the Results section (page 19-24).

- Motivation: Your motivation does not link with your research questions and analysis. If you want to check whether firms that had a scandal decide to engage more in CSR you should distinguish, in test, firms with a scandal in a given year and onward versus other firms.

>> Response: Once again, we are grateful to the reviewer for the constructive feedback. Additional literature and explanation on the problem of CSR reporting in the airline industry and the objective of this paper is added on page 4 in the Introduction section and page 10 in the Theoretical Background.

- Conclusions and implications: Poor job in providing implications of the research, for management, policy and stakeholders. Finally, in terms of conclusions, a general remark: Why is it important to focus on airline industry? What is specific to this context? What can we generalize to other industries?

>> Response: We thank the reviewer for this invaluable suggestion. We have rewritten the Contribution section with detailed examples to highlight the theoretical and practical implications for sustainable development and CSR reporting in the aviation industry (page 28-29).

Once again, we would like to take this opportunity to thank the Academic Editor and reviewers for their constructive and helpful comments.

---

## [Decision Letter · Decision Letter 1]

22 May 2020

Changing trends of corporate social responsibility reporting in the world-leading airlines

PONE-D-20-09210R1

Dear Dr. Yang,

We are pleased to inform you that your manuscript has been judged scientifically suitable for publication and will be formally accepted for publication once it complies with all outstanding technical requirements.

With kind regards,

Stefan Cristian Gherghina, PhD. Habil.

Academic Editor

PLOS ONE

Additional Editor Comments (optional):

Reviewers' comments:

Reviewer's Responses to Questions

**Comments to the Author**

1. If the authors have adequately addressed your comments raised in a previous round of review and you feel that this manuscript is now acceptable for publication, you may indicate that here to bypass the “Comments to the Author” section, enter your conflict of interest statement in the “Confidential to Editor” section, and submit your "Accept" recommendation.

Reviewer #1: All comments have been addressed

Reviewer #3: All comments have been addressed

2. Is the manuscript technically sound, and do the data support the conclusions?

Reviewer #1: Yes

Reviewer #3: (No Response)

3. Has the statistical analysis been performed appropriately and rigorously? 

Reviewer #1: Yes

Reviewer #3: (No Response)

4. Have the authors made all data underlying the findings in their manuscript fully available?

Reviewer #1: Yes

Reviewer #3: (No Response)

5. Is the manuscript presented in an intelligible fashion and written in standard English?

Reviewer #1: Yes

Reviewer #3: (No Response)

6. Review Comments to the Author

Reviewer #1: (No Response)

Reviewer #3: (No Response)

7. PLOS authors have the option to publish the peer review history of their article (what does this mean?). If published, this will include your full peer review and any attached files.

Reviewer #1: Yes: Kristina Marintseva

Reviewer #3: Yes: Tiago Gonçalves

ISEG Lisbon Shool of Economics & Management, Universidade de Lisboa

ADVANCE /CSG

---

## [Editor Report · Acceptance letter]

28 May 2020

PONE-D-20-09210R1 

Changing trends of corporate social responsibility reporting in the world-leading airlines 

Dear Dr. Yang:

I am pleased to inform you that your manuscript has been deemed suitable for publication in PLOS ONE. Congratulations! Your manuscript is now with our production department. 

With kind regards,

on behalf of

Dr. Stefan Cristian Gherghina 

Academic Editor

PLOS ONE